# Detecting newly installed bat boxes: Bats' prior familiarity with artificial roosts may play a bigger role than improved echo-reflective properties

**Anja Bergmann**[1,2]*, **Florian Gloza-Rausch**[1,3], **Mirjam Knörnschild**[1,3,4]*

**1** Museum für Naturkunde, Leibniz-Institute for Evolution and Biodiversity Science, Berlin, Germany, **2** Animal Behavior Lab, Free University of Berlin, Berlin, Germany, **3** Deutsche Fledermauswarte e.V., Berlin, Germany, **4** Evolutionary Ethology, Institute for Biology, Humboldt-Universität zu Berlin, Berlin, Germany

\* anja.bergmann@fu-berlin.de (AB); mirjam.knoernschild@mfn.berlin (MK)

## Abstract

Habitat loss in Europe severely affects bats, particularly tree-roosting species, due to the decreasing availability of tree cavities. One common conservation strategy is the installation of artificial roost boxes. However, the occupation of newly installed roost boxes can take up to several years, and the underlying mechanisms for successful roost detection in bats are still poorly understood. This study proposes enhancing the detectability of roost boxes to echolocating bats by incorporating hollow hemispheres that provide highly conspicuous echoes. The hemispheres strongly reflect the echolocation calls of passing bats and are thus well detectable over a broad range of angles. We hypothesized that roost boxes equipped with these hemispheres would attract more bats and exhibit greater bat activity than standard, unmodified boxes. To evaluate this, we placed 30 modified boxes and 30 unmodified boxes across three forest areas in Northern Germany, each differing in proximity to known bat hibernation sites and the prior presence of artificial roosts. We monitored bat activity by measuring light beam interruptions at each box and found that the activity of bats at the boxes varied considerably. Our findings indicate that, contrary to our hypothesis, bat activity was more strongly influenced by their prior experience with artificial roosts than by the increased detectability provided by hollow hemispheres. Furthermore, our study revealed that light beam interruptions indicated bat presence at the boxes earlier than visual checks for bats or feces, showcasing the benefits of non-invasive monitoring techniques. Conservation efforts are complex, and these results imply that for effective bat conservation, increasing bats' familiarity with artificial roosts may be more important than merely enhancing the detectability of these structures.

## Introduction

Bats represent one of Earth's most ecologically diverse and successful mammalian orders. They exhibit remarkable adaptations to various habitats, can actively fly, and orient via a sophisticated echolocation system. These adaptations are accompanied by specializations in

**Data availability statement:** The data underlying this study are available from the Harvard Dataverse repository (https://doi.org/10.7910/DVN/HCXRYR).

**Funding:** This work was supported by a stipend from the Elsa-Neumann Foundation and a follow-up funding by the Museum of Natural History Berlin to Anja Bergmann. Open Access Funding is provided by Freie Universität Berlin. The funders did not play a role in the study design, data collection and analysis, decision to publish, or preparation of the manuscript.

**Competing interests:** The authors have declared that no competing interests exist.

a broad variety of food sources and roosting opportunities. However, in the Anthropocene era, bat populations globally face severe threats due to habitat degradation, fragmentation, and loss. Bats' specialized life history strategy, comprising large aggregations for breeding and hibernation, coupled with low reproduction rates, renders them particularly susceptible to environmental changes [1].

In Germany, more than 50% of the occurring bat species are classified as endangered [2], and all are protected under European and National law [3]. Among the main drivers of their decline is the loss of natural habitats. Reasons for habitat loss may vary and include, but are not limited to deforestation or the early removal of old and dead trees. This leads to the critical loss of suitable tree cavities and thus roosting opportunities for tree-dwelling species [4,5]. While suitable trees are in decline worldwide [6], the development of cavities through natural processes is slow [7]. In response, artificial bat boxes have been installed, especially across Europe and North America, to mitigate the loss of natural roosting sites and to enlarge the habitats for endangered bat species [8]. Initially used to establish bat populations for pest control (as reported in [9] for instance), the use of artificial roosts clearly turned towards conserving endangered species in the past years (as reviewed in [10]).

Despite the potential benefits of artificial roosts, their effectiveness is often unpredictable due to an incomplete understanding of the specific requirements of target bat species, which can vary widely between different species and even among individuals. Factors affecting the success of roost boxes include box design [11–13], the number and deployment of boxes [14,15], the time since installation [16], and microclimate characteristics [17,18]. Moreover, the availability of other artificial roosting opportunities, such as bird boxes, also impacts bats' decisions regarding alternative habitats [15]. Consequently, while some roost boxes are in regular use after only a few days, other boxes will never be colonized.

Furthermore, how bats search for and find new roost sites is still poorly understood. For certain species, listening to the echolocation or social calls of conspecifics within the roost can reduce search time and increase detection rates [19–21]. Yet, additional olfactory cues such as guano or urine do not play a significant role in roost location [19,22,23]. Enhancing the detectability of bat boxes to passing bats could improve roost occupation, as locating roost entrances through echolocation presents a significant challenge.

One way could be to enhance the echo-reflective properties of newly installed boxes by incorporating an acoustic reflector. In the neotropics, certain tropical plants have evolved echo-reflective leaves or petals to attract nectar-feeding bats as pollinators [24,25]. These leaves or petals produce a distinctive, consistently strong echo across a broad range, serving as an echo-reflective cue. Such cues can be highly conspicuous to bats in an otherwise constantly changing environment, aiding them in locating plants and guiding them to nectar sources, thereby increasing the likelihood of successful pollination [24]. Hollow hemispheres hold the potential to replicate similarly characteristic echo patterns, resembling the bell-shaped concave form of bat-pollinated flower petals or leaves. Nectar-feeding bats can distinguish between hemispheres of different sizes [25], and in experimental settings, temperate zone bats can learn to associate them with suitable roosts [26]. This approach holds the potential for improving the detectability of artificial roosts, potentially facilitating their quicker and more successful adoption by bats.

We aimed to assess the effectiveness of hollow hemispheres as cues to improve the detectability of newly installed bat boxes and thereby enhance occupancy rates. Therefore, gable boxes were equipped with hollow hemispheres (modified boxes) and installed alongside unmodified boxes in three distinct forest sites that are known summer habitats of various *Myotis* species and near popular hibernacula (winter roosts). Anticipating higher detectability,

we hypothesized increased activity rates at modified boxes compared to unmodified boxes. We furthermore assumed that bats with prior exposure to hollow hemispheres at their hibernaculum would be more likely to settle in modified boxes compared to bats without such exposure. Those assumptions were tested by comparing the activity rates of bats at boxes with and without hollow hemispheres across the three study sites.

## Methods

### Pre-observation

To ensure that bats were not deterred by the hollow hemispheres, we conducted a pre-observation at the Kalkberg Cave, a large hibernaculum in Northern Germany (Bad Segeberg, 10°18′57″E, 53°56′09″N), hosting more than 30,000 hibernating bats each winter. The vicinity of the cave is extensively used for late summer and autumn swarming prior to hibernation.

We tested a configuration of five hollow stainless-steel hemispheres, each with a diameter of 10 cm, arranged in a row and mounted on a wooden plank at a blind side entrance without a connection to the cave in the 2018 swarming season. Using a night-vision camera, bat activity at the entrance was observed for 10 minutes, counting the number of bats within a 50 cm radius around the hemispheres. We compared a total of eight observations with and without hollow hemispheres, respectively using a Wilcoxon test (SPSS, version 20, SPSS Inc., Chicago, IL, USA). Subsequently, in autumn 2019, we installed hollow hemispheres at the two main entrances of the cave to habituate incoming and departing bats to the specific echoes of the hemispheres and allow the bats to associate the hemispheres with a suitable hibernaculum. The hemispheres remained at the entrances until spring 2021.

### Box set up and equipment

We utilized gable boxes of the type FLH14 with a 14 mm opening (27 × 18 × 25 cm, Hasselfeldt GmbH, Aukrug, Germany) and attached the hemispheres with a diameter of 10 cm to the front part of the boxes' roof with a screw (modified boxes, Fig 1). All boxes were then equipped with self-made light barrier systems, based on an Arduino circuitry. Sender and receiver units were affixed on both sides of the box entrance. When the connection between both LEDs was interrupted, the event was recorded on an SD card. The conductor board with the computer and the electrical power supply were stored in a waterproof case attached to the sidewall of the bat box. Regular maintenance involved replacing batteries every three weeks and visually inspecting boxes for bats and/or feces.

### Study sites

The study was carried out across three forest sites in Schleswig-Holstein, Germany. In each forest, we installed ten modified boxes (with hollow hemispheres) and ten unmodified boxes (without hollow hemispheres). Installation was conducted in March (Forest A and C) and April 2021 (Forest B), with observations continuing until September 2021.

Forest A is located about 8 km from Kalkberg Cave, Forest B is located 21.5 km further north (Fig 2). Bats hibernating in the Kalkberg Cave roosted in both forests during the summer. As both forests had been equipped with artificial bat boxes before our study, we assume that bats roosting in those forests are already familiar with bat boxes. Forest C is about 66 km away from the Kalkberg Cave, thus bats were not habituated to the hemispheres, but it is located in the vicinity of another mass hibernaculum of *Myotis* bats (MUNA-Kropp: 9°52′47″E, 54°39′75″N). In contrast to Forest A and B, no boxes had been present prior to our study.

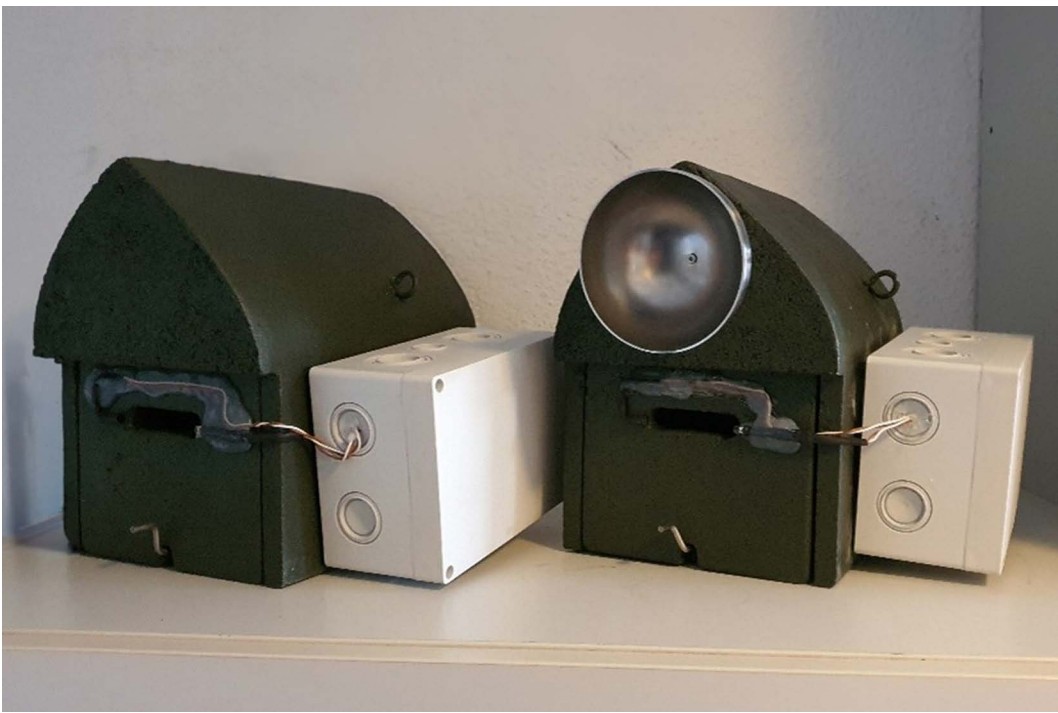

**Fig 1. Unmodified and modified artificial roosts.** Bat activity was compared between gable boxes without and with an attached hollow hemisphere as an echo-reflective cue. Activity was measured as light beam interruptions by the diodes affixed to both sides of the entrances. The waterproof box on the side houses the light beam system's electrical components and power supply.

The study was carried out within the framework of the species conservation exemption permit issued by the LLUR 512 Schleswig-Holstein on 07/03/2018.

## Data analysis

To determine whether the activity at boxes with an echo-reflective cue differed from that without such a cue, we conducted a comparison based on the number of light-beam interruptions observed on modified and unmodified boxes. We specifically focused on interruptions between 7 pm and 7 am, as these were most likely associated with bat activity at the box. For the analysis, interruptions with a duration ranging from 1 second to 15 seconds were considered, while those exceeding 15 seconds were excluded. This exclusion was necessary as it was not possible to determine whether such interruptions were caused by bats blocking the entrance, any technical issues, or other animals.

Particularly at the beginning of the observation period, we encountered some technical issues related to light beam or SD card failure. These issues may have arisen due to extremely cold nights during March and April. As the season progressed, certain boxes were occupied by species other than bats, especially hornets, rendering them uninhabitable for bats. To account for the fluctuating number of operational boxes over time, interruptions per box-night (I/BN) was calculated as a relative measure of activity. Box nights were determined by counting nights during which the light beams were operational, and boxes were confirmed as unoccupied by species other than bats. The total number of interruptions was then divided by box-nights for each box individually. The total number of observable nights per study site was 193 at both sites A and C, and 171 at site B, resulting in a maximum total of 3,860 box-nights

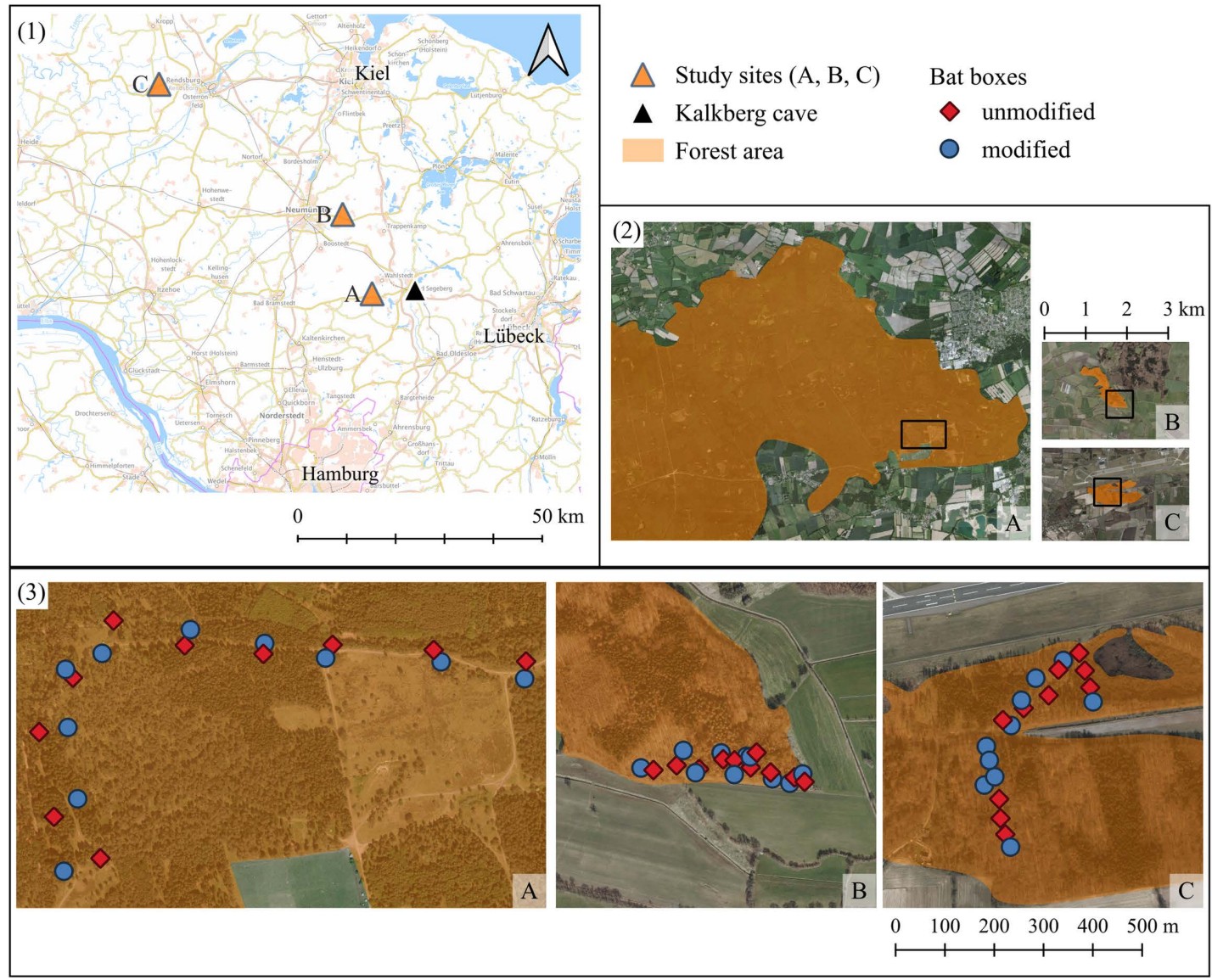

**Fig 2. Study sites and box locations.** (1) Overview of the three study sites in Schleswig-Holstein, Germany. (2) Shapes and relative sizes of the sites. Rectangles mark the sections shown in (3), with detailed bat box positions within the three forests. Modified bat boxes had a hollow hemisphere attached to the front roof as an acoustic reflector. Scales: (1) 1:2,000,000; (2) 1:240,000; (3) 1:20,000; Sources: (1) © Bundesamt für Kartographie und Geodäsie (2024). https://sgx.geodatenzentrum.de/web_public/Datenquellen_TopPlus_Open.pdf. Licensed under Creative Commons Attribution 4.0 International (CC BY 4.0); (2,3) © GeoBasis-DE/LVermGeo SH/CC BY 4.0. Licensed under Creative Commons Attribution 4.0 International (CC BY 4.0).

for sites A and C, and 3,420 for site B. The actual box-nights varied due to the aforementioned occupancy by other species or technical problems, amounting to 2,598 box-nights for site A, 2,135 for site B, and 1,877 for site C (Table 1).

To assess which factors affected bat activity at the boxes (measured as I/BN), we employed a generalized linear model (GLM) with a Gamma family and log link. The model included I/BN as the dependent variable and modification and location as fixed factors. The significance of the predictors was evaluated using a Wald Chi-squared test and subsequent Tukey's post hoc tests (R v. 4.4.1, The R Foundation for Statistical Computing).

## Results

### Hemispheres do not deter bats

The pre-observation at the blind side entrances of the Kalkberg Cave did not reveal any difference in bat activity when the hollow hemispheres were present or absent (Wilcoxon-Test: $Z = -0.560$, $N = 8$, exact $p = 0.641$). Consequently, we concluded that the hollow hemispheres did not have any deterrent impact on the swarming bats and proceeded with attaching hemispheres to the main entrances of the Kalkberg Cave and modifying artificial roosts.

### Bat activity at boxes varied considerably

The bat activity at the boxes varied considerably, with an overall mean of 9.03 interruptions per box-night (I/BN) for unmodified boxes and a mean of 4.87 I/BN for modified boxes (Table 2).

The activity level was higher at boxes at sites A and B, where artificial roosts had been present before our study (Fig 3). Unmodified boxes in Forests A and B displayed the highest bat activity, while the lowest activity was observed at the modified boxes in Forests B and C. However, in Forest C, the activity was comparatively similar between treatments.

A Generalized linear model (GLM) was fitted with I/BN as dependent variable, modification and location as fixed factors, and Gamma family with log link. The analysis indicated that the I/BN did not change with modification when observed across all sites (estimate = 0.5280, t-value = 1.9155, p = 0.0609).

Compared to study site C, I/BN was higher at site A (estimate = 0.7311, t-value = 2.085, p = 0.0334) and B (estimate = 0.7068, t-value = 2.185, p = 0.0420). However, these results did not remain significant after post hoc testing, revealing that differences between study sites were not statistically significant (Tukey's post hoc test: A – B: estimate = -0.0244, t = -0.0718, p = 0.9972; A – C: estimate = -0.7311, t = -2.1848, p = 0.0833; B – C: estimate = -0.7068, t = -2.0847, p = 0.1029).

**Table 1. Characterizations of the three study sites.**

| Study Site | Summer roost of bats hibernating in the Kalkberg Cave | Prior familiarity with boxes | Maximum possible number of | | Actual number of box-nights |
|---|---|---|---|---|---|
| | | | Observable nights per box | Box-nights per forest | |
| A | Yes | Yes | 193 | 3,860 | 2,598 |
| B | Yes (Potentially) | Yes | 171 | 3,420 | 2,135 |
| C | No | No | 193 | 3,860 | 1,877 |

**Table 2. Number of interruptions and box nights per forest for modified and unmodified boxes.**

| | | Unmodified boxes | | | Modified boxes | | |
|---|---|---|---|---|---|---|---|
| | | Interruptions | Box nights | I/BN | Interruptions | Box nights | I/BN |
| Study site | A | 15,488 | 1,490 | **9.52** | 8,877 | 1,108 | **6.95** |
| | B | 13,308 | 1,142 | **13.33** | 4,150 | 993 | **4.32** |
| | C | 3,554 | 811 | **4.19** | 4,094 | 1,066 | **3.49** |
| | total | 32,350 | 3,443 | **9.03** | 17,121 | 3,167 | **4.87** |

I/BN = number of interruptions per box night, calculated as the mean I/BN over all boxes per study site.

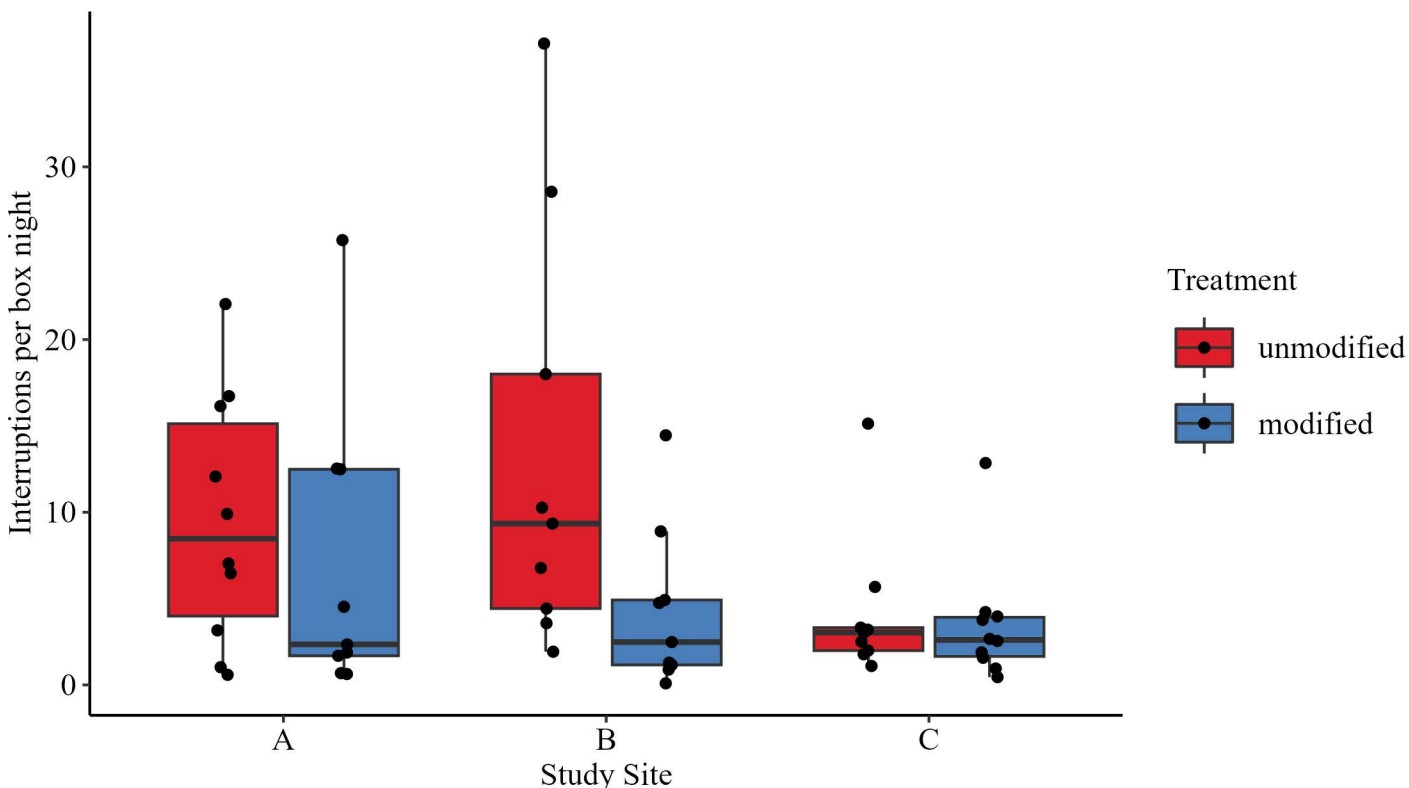

**Fig 3. Distribution of interruptions per box-night (I/BN) by treatment and location.** The I/BN differed depending on box modification with hollow hemispheres and the prior familiarity of bats with bat boxes. Prior familiarity of bats was given on study sites A and B.

### Light beams detected bat activity earlier than visual inspections did

In addition to the automated monitoring via light beams, the boxes were visually controlled every three weeks between April and September. On average, the first light beam interruptions could be observed after about a month, while the first sighting of individuals or feces took more than 100 days on average (Table 3, Fig 4).

## Discussion

We aimed to investigate whether hollow hemispheres, as highly conspicuous echo-reflective cues, influence bat activity at newly installed bat boxes. By comparing the mean number of light beam interruptions per night, we found that the activity of bats at the boxes varied considerably. However, the results suggest that the primary factor influencing activity was not the presence of hollow hemispheres at the boxes, but rather the bats' familiarity with boxes as artificial roosting opportunities within the surveyed forest sites.

Notable differences in activity were detected between Forests A and B compared to Forest C, with overall activity levels being lower in Forest C. However, this trend was not statistically significant in post-hoc analyses. Bats roosting in Forests A and B are likely to hibernate in the Kalkberg Cave, where they presumably encountered hollow hemispheres during the swarming period. Thus, at first glance, the differences in activity rates could originate from prior exposure to hollow hemispheres as hypothesized. However, at sites A and B the activity was much higher at unmodified boxes, suggesting that prior exposure to the hemispheres does not positively influence bat activity.

**Table 3. Mean number of days after which the first light beam interruption, feces, or bats were observed at the boxes.**

| | | Unmodified boxes | | | Modified boxes | | |
|---|---|---|---|---|---|---|---|
| | | Light beam | Feces | Bats | Light beam | Feces | Bats |
| **Study site** | A | **36.4** (10)<br>7-68 | **136.33** (6)<br>104-193 | **141** (6)<br>104-173 | **43.33** (9)<br>16-68 | **148.5** (4)<br>104-193 | **156.67** (3)<br>104-193 |
| | B | **40.2** (10)<br>26-82 | **122.75** (8)<br>81-171 | **125.33** (3)<br>82-171 | **36.75** (8)<br>28-91 | **107.25** (4)<br>81-123 | **127.33** (3)<br>82-150 |
| | C | **45.11** (9)<br>6-86 | **145** (1) | (0) | **45.6** (10)<br>3-87 | **78.5** (2)<br>33-124 | **124** (1) |

The total number of boxes in which the respective observation was made is given in parenthesis, with a maximum of 10 boxes per treatment. The second row shows the minimum and maximum number of days until the first observation per box.

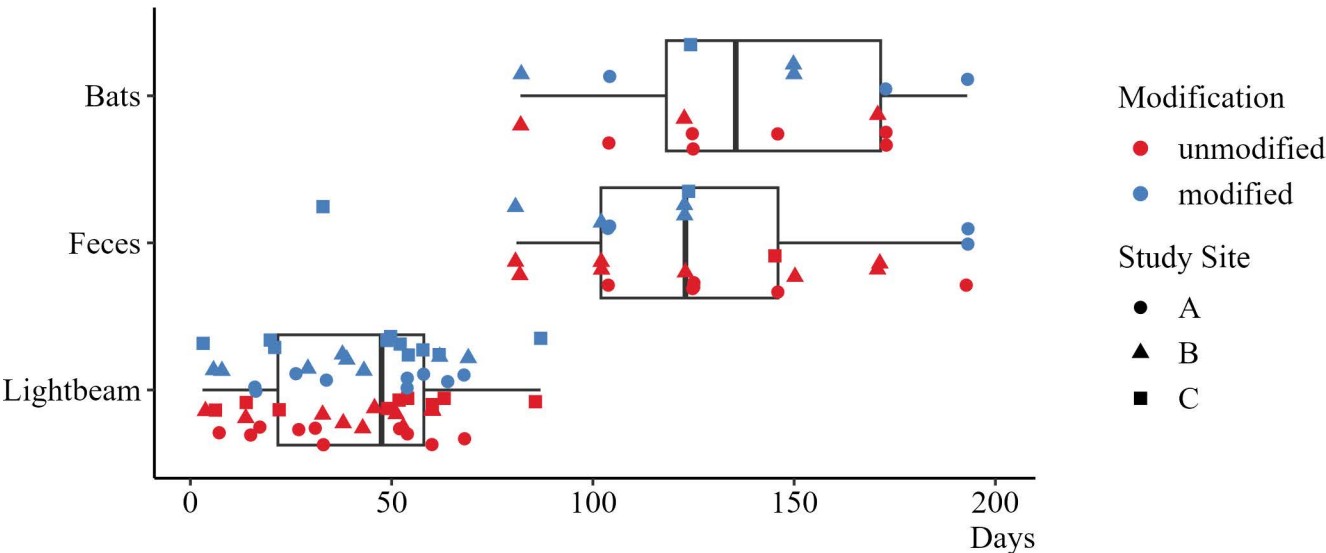

**Fig 4. Comparison between observation methods for bat activity.** At both modified and unmodified boxes, light beam interruptions indicated bat activity several days or weeks before feces or bats were detected during visual inspections.

Hernández-Montero et al. [26] demonstrated that while bats are able to differentiate between suitable and unsuitable roosts based on size variations in attached hollow hemispheres, they nevertheless revisit all boxes in subsequent seasons. This behavior may reflect either a lack of memory retention during hibernation or a necessity to reassess potential roosting opportunities that may have changed over time. Consequently, pre-exposure to hollow hemispheres appears to lack the anticipated effect on bat activity.

The second key difference between sites A and B compared to C was that bat boxes as artificial roosting opportunities had been in use previously in Forests A and B, while no such structures existed in Forest C before. Consequently, at site C, where bats lacked prior familiarity with the boxes, both overall activity and the differences between modified and unmodified boxes were notably lower. Finding new roosts can be very time-consuming for bats, especially if they are not already using similar structures nearby. It has been observed before that the detection of newly installed boxes becomes more likely if bats are familiar with boxes [15]. Given the presence of boxes in Forests A and B, it is reasonable to infer that bats roosting in

these areas are accustomed to them, as bats are known to share information about suitable roosting opportunities among conspecifics [27]. As there were no artificial roosts at site C prior to our study, bats may need additional time to detect, habitually occupy, and use them. The similar activity rates observed between modified and unmodified boxes at site C further emphasize that the presence of hollow hemispheres did not influence activity rates.

The observed trend of higher activity at unmodified boxes at both sites A and B might result from bats familiar with artificial roosts having a predefined "search pattern" to identify suitable roost characteristics. The introduction of hollow hemispheres could disrupt this pattern, causing these bats to inspect new structures less frequently. When comparing the two sites, bat activity at unmodified boxes was approximately similar, whereas activity at modified boxes was comparatively higher at site A. At site B, the differences in activity rates between box types were more pronounced. Given that site B is a smaller forest with a high density of roosts, bats may face less difficulty in locating new roosts and may exhibit reduced interest in novel structures like the modified boxes. Increased box density in an area reduces occupancy rates, as a fixed number of bats spread over a higher number of boxes. The overall box occupancy rates typically begin to rise gradually when bats start to reproduce [13,28]. From this perspective, activity at modified boxes could potentially increase over time. The lower overall roost saturation at site A and a more widespread setup across the forest may mitigate these effects, leading to less pronounced differences between modified and unmodified boxes at site A.

Taken together, the hollow hemispheres did not lead to an increase in bat activity at modified boxes. Considering the higher activity at unmodified boxes, bats seem to prefer them. Additional factors contributing to this preference may include neophobia, as bats may exhibit hesitance or disinterest toward unfamiliar structures. Bats accustomed to artificial roosts and regularly using them already likely recognize the characteristics they seek in a suitable roost. Hence, they might not need to adjust to the altered appearance of the modified boxes, especially if sufficient unmodified boxes or other roosting opportunities exist in the habitat. Interestingly, activity levels were nearly identical between treatments in forest C, where bats lacked familiarity with boxes, further supporting the influence of established search patterns.

We assumed that attaching hemispheres to boxes would enhance their detectability. Unfortunately, due to technical issues arising after the installation of the boxes, we cannot confidently determine whether the modified boxes were located earlier than the unmodified ones. Consequently, we are unable to conclude the efficacy of the hemispheres in improving box detectability. However, a comparable approach found that echo-reflective cues did not consistently reduce search time, and contrary to their expectations, the reflectors did not significantly enhance detectability on a measurable level [29]. In summary, despite reflecting a conspicuous echo, the hollow hemispheres might have altered the boxes so that bats no longer recognize them as suitable roosts.

Besides, we would like to highlight our monitoring technique. Despite initial technical challenges, the light beams operated reliably, revealing high levels of activity at some boxes, even though we never visually observed a single bat during our controls. Some bat species are known to change their roosts nightly, making it challenging to conclude the use of a specific roost box during such monitoring. By checking boxes once per season, Pschonny et al. [13] found 40% of newly installed gable boxes were occupied within the first year. However, different observation methods can yield varying results, and feces may naturally be found in more boxes than bat individuals [30]. Our light beam systems registered interruptions indicating bat activity at over 90% of the boxes. The majority of interruptions occurred long before any individuals or feces were visually confirmed inside boxes, making it a suitable method to measure bat activity.

In conclusion, automated monitoring systems, such as light beams, offer a valuable solution to assess bat activity in detail. These low-cost systems enable non-invasive monitoring over extended periods. However, we recommend conducting a trial of the final set-up on a limited number of systems for at least one season to address potential technical issues proactively. By the end of our study period, we observed bat activity at over 90% of the newly installed boxes. It appears that at least in this study area, bats did not encounter significant challenges in locating new roosts. Therefore, providing a substantial number of artificial roosting opportunities could be a beneficial complement to the ongoing efforts to preserve natural habitats.

## Acknowledgments

First of all, we would like to thank Andreas Rose for sharing the prototype of the light beam with us and for ongoing support with upcoming technical issues. Furthermore, we are especially grateful to Sebastian Bergmann, Juliane Lukas and Anka Bergmann for their invaluable help, field support, and assistance with soldering.

Our sincere thanks go to Matthias Göttsche from Bat Monitoring of the Federal State Schleswig-Holstein and the Noctalis team for granting us access to the Kalkberg Cave and providing logistical support. Special appreciation is extended to Helmut Giljum (Noctalis) for reattaching falling hemispheres at the cave entrances. We would like to thank the Bundesanstalt für Immobilienaufgaben, Bundesforstbetrieb Trave, and the responsible district foresters Svenja Weihe and Bernd von Kamptz for their kind support in selecting the locations for the installation of the bat boxes in Forests A and C. We would also like to thank Dr. Björn Rickert and his team from the regional group of NABU Neumünster very much for their kind permission to carry out the study in Forest B and their interest in our work.

## Author contributions

**Conceptualization:** Anja Bergmann, Florian Gloza-Rausch, Mirjam Knörnschild.

**Data curation:** Anja Bergmann.

**Formal analysis:** Anja Bergmann.

**Funding acquisition:** Anja Bergmann.

**Investigation:** Anja Bergmann.

**Methodology:** Anja Bergmann, Florian Gloza-Rausch, Mirjam Knörnschild.

**Project administration:** Anja Bergmann, Florian Gloza-Rausch, Mirjam Knörnschild.

**Resources:** Florian Gloza-Rausch, Mirjam Knörnschild.

**Supervision:** Florian Gloza-Rausch, Mirjam Knörnschild.

**Validation:** Florian Gloza-Rausch, Mirjam Knörnschild.

**Visualization:** Anja Bergmann.

**Writing – original draft:** Anja Bergmann.

**Writing – review & editing:** Anja Bergmann, Florian Gloza-Rausch, Mirjam Knörnschild.

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
