## [Decision Letter · Decision Letter 0]

15 Jul 2024

PONE-D-24-23204Detecting newly installed bat boxes: Bats’ prior familiarity with artificial roosts plays a bigger role than improved echo-reflective propertiesPLOS ONE

Dear Dr. Bergmann,

Thank you for submitting your manuscript to PLOS ONE. After careful consideration, we feel that it has merit but does not fully meet PLOS ONE’s publication criteria as it currently stands. Therefore, we invite you to submit a revised version of the manuscript that addresses the points raised during the review process.

We look forward to receiving your revised manuscript.

Kind regards,

Laith Al-Eitan

Academic Editor

PLOS ONE

3. We note that Figure 1 in your submission contain copyrighted images. All PLOS content is published under the Creative Commons Attribution License (CC BY 4.0), which means that the manuscript, images, and Supporting Information files will be freely available online, and any third party is permitted to access, download, copy, distribute, and use these materials in any way, even commercially, with proper attribution. For more information, see our copyright guidelines: http://journals.plos.org/plosone/s/licenses-and-copyright.

Reviewers' comments:

Reviewer's Responses to Questions

**Comments to the Author**

1. Is the manuscript technically sound, and do the data support the conclusions?

Reviewer #1: Partly

Reviewer #2: Yes

2. Has the statistical analysis been performed appropriately and rigorously?

Reviewer #1: No

Reviewer #2: Yes

3. Have the authors made all data underlying the findings in their manuscript fully available?

Reviewer #1: Yes

Reviewer #2: Yes

4. Is the manuscript presented in an intelligible fashion and written in standard English?

Reviewer #1: Yes

Reviewer #2: Yes

5. Review Comments to the Author

Reviewer #1: PONE-D-24-23204 summarizes a study on whether modifications (adding hollow hemispheres) of bat boxes could increase the chance of bats using/occupying a bat box. The experimental design and execution are generally sound. I believe authors have collected valid and valuable data. However, I do think that the statistical analysis must be improved, and therefore some of the current conclusions are somewhat farfetched based on the analysis and should be revised. I recommend a major revision for the manuscript.

In terms of statistical analysis, I cannot understand why authors used “study site” as a random effect. Authors already included “familiarity with boxes” as a dependent variable to investigate its significance. It is important to note that “familiarity with boxes” is a variable nested within “study site” (see Table 1). Therefore I do not think it is appropriate. From the analytical perspective, it makes more sense just using “study site” as a dependent variable instead of a random effect and not including “familiarity with boxes”. From the biological perspective, there is no proof that individuals that used/occupied bat boxes at site A and B had any prior familiarity with boxes. There is a difference between a population being exposed to something versus all individuals in a population being exposed to something. Additionally, I do not understand how the authors came up with the statement in Line 130-131. The only thing I could find is Line 91-94 and Line 106-109 where authors talked about exposing bats to hollow hemispheres at site A and possibly B (not the actual bat boxes based on the writing).

Based on what authors presented, it seems that at two sites (A and B) bats preferred unmodified boxes over modified boxes whereas no difference was found at site C. The peculiar part is that bats at site A and B were previously exposed to the modification/ hollow hemispheres. Overall it reads like a negative result to me. Personally I think it is very important to publish and share negative results from research, instead of treating them like some kind of stigma. However, I really cannot understand how authors had a conclusion on prior familiarity with artificial roosts as highlighted in the title. This part is either poorly explained/presented or a farfetched conclusion. My suggestion is: present the differences among sites and speculate what aspects of those sites might contribute to the results. A few things I noticed from the map include: 1) surrounding landscape differences, 2) forest patch size differences, 3) the layout of bat boxes (site A and B had this one on one “matched” layout whereas C had a more clustered layout). Unless there is something missing in the manuscript, I would strongly recommend toning down on the prior familiarity aspect.

Additionally, I found some word choices strange. For example, authors used “visibility” to describe bat boxes. I suppose we still do not have a lot of research on how bats find and use bat boxes and the terminology isn’t mature. But I think most researchers would automatically think visibility is related to see/be seen. Maybe detectability is a better word (?) as clearly the hollow hemispheres were designed to attract bats acoustically not visually. Another example is “occupancy”, as it is so heavily used in occupancy models. I admit that I do not have a good suggestion for authors (maybe usage).

I also suggest revising figure 4 (to separate sites like figure 3) and Table 2 (part of the first column missing on my end).

Overall, I look forward to seeing this work being published after revisions.

Reviewer #2: Comments

Introduction section

• The reference “Altringham, 2011” should be numbered and added to the list of references.

• Paragraph starting with line 88-95 should be incorporated with the methods section, and place the assumptions in the last paragraph starting with line 82.

Authors should clarify how bats become “familiar” with the boxes.

Authors should indicate the species of bats that occur within the study area, and if there are variations among species in selecting the installed boxes.

6. PLOS authors have the option to publish the peer review history of their article (what does this mean? ). If published, this will include your full peer review and any attached files.

**Do you want your identity to be public for this peer review?** For information about this choice, including consent withdrawal, please see our Privacy Policy .

Reviewer #1: **Yes: ** Han Li

Reviewer #2: **Yes: ** Zuhair S. Amr

---

## [Author Response · Author response to Decision Letter 1]

4 Oct 2024

Journal Requirements:

1. Copyright Issues:

o We provided a replacement for Figure 2 that complies with the CC BY 4.0 license. We added all the necessary license information in the figure caption.

2. Manuscript Formatting:

o We have ensured that the manuscript adheres to PLOS ONE’s formatting requirements as specified.

Reviewer #1:

o Comment: In terms of statistical analysis, I cannot understand why authors used “study site” as a random effect. Authors already included “familiarity with boxes” as a dependent variable to investigate its significance. It is important to note that “familiarity with boxes” is a variable nested within “study site” (see Table 1). Therefore I do not think it is appropriate. From the analytical perspective, it makes more sense just using “study site” as a dependent variable instead of a random effect and not including “familiarity with boxes”.

o Response: Thank you very much for your insightful feedback regarding our statistical analysis. We have revised the model accordingly and changed it to a Generalized Linear Model (GLM), which indeed provides a better fit. As per your suggestion, we now include "study site" in addition to "modification" as dependent variables, and we have removed "familiarity with boxes" to avoid the nesting issue. The updated results and analysis can be found from line 219 (version with track changes) onwards.

o Comment: From the biological perspective, there is no proof that individuals that used/occupied bat boxes at site A and B had any prior familiarity with boxes. There is a difference between a population being exposed to something versus all individuals in a population being exposed to something. Additionally, I do not understand how the authors came up with the statement in Line 130-131. The only thing I could find is Line 91-94 and Line 106-109 where authors talked about exposing bats to hollow hemispheres at site A and possibly B (not the actual bat boxes based on the writing).

o Response: The statement refers to the prior familiarity of bats with roost boxes in forests A and B. Since both forests had been equipped with artificial bat boxes prior to our study, we assume that bats roosting in those forests are also familiar with such structures, as information transfer between colony members has been observed in the past (also see revised part of the discussion from line 264 onwards). However, we understand that this assumption was not made sufficiently clear in the original text. To address this, we have further revised the method section and added the following: “As both forests had been equipped with artificial bat boxes before, we assume that bats roosting in those forests are already familiar with bat boxes.” in line 144. We have also added a similar clarification for forest C, where prior exposure to bat boxes is less likely, in line 146.

o Comment: Based on what authors presented, it seems that at two sites (A and B) bats preferred unmodified boxes over modified boxes whereas no difference was found at site C. The peculiar part is that bats at site A and B were previously exposed to the modification/ hollow hemispheres. Overall it reads like a negative result to me. Personally I think it is very important to publish and share negative results from research, instead of treating them like some kind of stigma. However, I really cannot understand how authors had a conclusion on prior familiarity with artificial roosts as highlighted in the title. This part is either poorly explained/presented or a farfetched conclusion. My suggestion is: present the differences among sites and speculate what aspects of those sites might contribute to the results. A few things I noticed from the map include: 1) surrounding landscape differences, 2) forest patch size differences, 3) the layout of bat boxes (site A and B had this one on one “matched” layout whereas C had a more clustered layout). Unless there is something missing in the manuscript, I would strongly recommend toning down on the prior familiarity aspect.

o Response: We have revised the discussion to clarify our focus on prior familiarity with artificial roosts. Despite the site differences, based on a review of the literature we still believe that prior familiarity remains an important factor influencing bat behavior regarding roost occupation rate and timing. However, given the new results of the GLM, we now acknowledge that this effect appears more as a tendency rather than a statistically significant outcome. We hope the revised discussion provides a more balanced interpretation of the results and better explains how we arrived at this conclusion. We also revised the title accordingly.

o Comment: Additionally, I found some word choices strange. For example, authors used “visibility” to describe bat boxes. I suppose we still do not have a lot of research on how bats find and use bat boxes and the terminology isn’t mature. But I think most researchers would automatically think visibility is related to see/be seen. Maybe detectability is a better word (?) as clearly the hollow hemispheres were designed to attract bats acoustically not visually. Another example is “occupancy”, as it is so heavily used in occupancy models. I admit that I do not have a good suggestion for authors (maybe usage).

o Response: We are grateful for your suggestions and have carefully revised the manuscript to improve clarity and consistency in terminology. As for the use of the term "visibility," we agree that "detectability" is a more precise term in the context of our study, given that the hollow hemispheres were designed to enhance acoustic detection by bats rather than visual visibility. We have revised the manuscript to replace "visibility" with "detectability" accordingly. However, regarding the term "occupation," we would prefer to maintain this wording as we believe it best captures the active process by which bats start utilizing the roost boxes over time. It is furthermore used in the literature quite frequently, so we hope there will be no misinterpretation.

o Comment: I also suggest revising figure 4 (to separate sites like figure 3) and Table 2 (part of the first column missing on my end).

o Response: Thank you for your suggestion regarding Figure 4. In response, we have added different symbols to depict the sites, making the data from each site distinguishable. We chose not to fully separate the sites as in Figure 3, as this would have added unnecessary complexity to the figure without significantly enhancing the clarity of the data. We believe the updated version strikes a balance between clarity and simplicity. Additionally, we have corrected Table 2, which appears to have been a formatting issue, and it should now display properly on a single page.

Reviewer #2:

o Comment: Reference “Altringham, 2011” should be numbered and added.

o Response: Thanks for pointing this out, the reference has now been numbered and added to the reference list.

o Comment: Paragraph starting with line 88-95 should be incorporated with the methods section, and place the assumptions in the last paragraph starting with line 82.

o Response: We have revised the last paragraph of the introduction for clarity. Most of the information was already present in the methods section, but we have restructured the assumptions from line 84 (track change version) onwards.

o Comment: Authors should clarify how bats become “familiar” with the boxes.

o Response: Thank you for your comment. In response, and in line with Reviewer 1's feedback, we have clarified how we define "familiarity" with the boxes in the methods section, starting from line 143.

o Comment: Authors should indicate the species of bats that occur within the study area, and if there are variations among species in selecting the installed boxes.

o Response: Across Schleswig-Holstein, 15 bat species are known to occur (see [2] Meining et al.). However, we cannot definitively identify which species utilized the boxes, as bats were only sporadically encountered during visual checks. Furthermore, as shown in Figure 4, far more activity was recorded through light beam interruptions than via visual controls, making species identification based on these observations incomplete and unreliable. Thus, unfortunately, we cannot provide reliable information concerning species composition in the study areas or the boxes.

---

## [Decision Letter · Decision Letter 1]

2 Mar 2025

Detecting newly installed bat boxes: Bats’ prior familiarity with artificial roosts may play a bigger role than improved echo-reflective properties

PONE-D-24-23204R1

Dear Dr. Anja,

We’re pleased to inform you that your manuscript has been judged scientifically suitable for publication and will be formally accepted for publication once it meets all outstanding technical requirements.

Kind regards,

Laith Naser Al-Eitan, Ph.D

Academic Editor

PLOS ONE

Additional Editor Comments (optional):

Reviewers' comments:

Reviewer's Responses to Questions

**Comments to the Author**

1. If the authors have adequately addressed your comments raised in a previous round of review and you feel that this manuscript is now acceptable for publication, you may indicate that here to bypass the “Comments to the Author” section, enter your conflict of interest statement in the “Confidential to Editor” section, and submit your "Accept" recommendation.

Reviewer #2: All comments have been addressed

2. Is the manuscript technically sound, and do the data support the conclusions?

Reviewer #2: Yes

3. Has the statistical analysis been performed appropriately and rigorously?

Reviewer #2: Yes

4. Have the authors made all data underlying the findings in their manuscript fully available?

Reviewer #2: Yes

5. Is the manuscript presented in an intelligible fashion and written in standard English?

Reviewer #2: Yes

6. Review Comments to the Author

Reviewer #2: The authors responded to all comments. I think the manuscript is much better format by now and meets the standards of the Journal.

7. PLOS authors have the option to publish the peer review history of their article (what does this mean? ). If published, this will include your full peer review and any attached files.

**Do you want your identity to be public for this peer review?** For information about this choice, including consent withdrawal, please see our Privacy Policy .

Reviewer #2: **Yes: ** Zuhair Amr

---

## [Editor Report · Acceptance letter]

PONE-D-24-23204R1

PLOS ONE

Dear Dr. Bergmann,

I'm pleased to inform you that your manuscript has been deemed suitable for publication in PLOS ONE. Congratulations! Your manuscript is now being handed over to our production team.

Kind regards,

on behalf of

Professor Laith Naser Al-Eitan

Academic Editor

PLOS ONE